# Deciphering Genetic Alterations of Taiwanese Patients with Pancreatic Adenocarcinoma through Targeted Sequencing

**DOI:** 10.3390/ijms23031579

**Published:** 2022-01-29

**Authors:** Chi-Cheng Huang, Chih-Yi Liu, Chi-Jung Huang, Yao-Chun Hsu, Heng-Hui Lien, Jia-Uei Wong, Feng-Chuan Tai, Wen-Hui Ku, Chi-Feng Hung, Jaw-Town Lin, Ching-Shui Huang, Han-Sun Chiang

**Affiliations:** 1Division of General Surgery, Department of Surgery, Taipei Veterans General Hospital, Taipei 11217, Taiwan; chishenh74@gmail.com; 2Comprehensive Breast Health Center, Taipei Veterans General Hospital, Taipei 11217, Taiwan; 3Institute of Epidemiology and Preventive Medicine, College of Public Health, National Taiwan University, Taipei 100, Taiwan; 4Department of Pathology, Cathay General Hospital SiJhih, New Taipei 221, Taiwan; cyl1124@gmail.com; 5Department of Medical Research, Cathay General Hospital, Taipei 106, Taiwan; science.man2@gmail.com; 6Department of Biochemistry, National Defense Medical Center, Taipei 114, Taiwan; 7Division of Gastroenterology, Department of Internal Medicine, E-da Hospital, Kaohsiung 82445, Taiwan; holdenhsu@gmail.com; 8Division of General Surgery, Department of Surgery, Cathay General Hospital, Taipei 106, Taiwan; hhlhhl@cgh.org.tw (H.-H.L.); cghsurgery@gmail.com (F.-C.T.); 9School of Medicine, College of Medicine, Fu-Jen Catholic University, New Taipei 242, Taiwan; skin@mail.fju.edu.tw; 10Division of General Surgery, Department of Surgery, Fu-Jen Catholic University Hospital, New Taipei 243, Taiwan; drjuboy@gmail.com; 11Department of Clinical Pathology and Molecular Medicine, Taipei Institute of Pathology, Taipei 10374, Taiwan; whku0322@gmail.com; 12Digestive Medicine Center, China Medical University Hospital, Taichung 404, Taiwan; jawtown@gmail.com; 13School of Medicine, College of Medicine, Taipei Medical University, Taipei 110, Taiwan

**Keywords:** pancreatic adenocarcinoma, next-generation sequencing, targeted sequencing, Taiwan, actionable mutation

## Abstract

Pancreatic adenocarcinoma (PAC) is the 8th leading cause of cancer-related deaths in Taiwan, and its incidence is increasing. The development of PAC involves successive accumulation of multiple genetic alterations. Understanding the molecular pathogenesis and heterogeneity of PAC may facilitate personalized treatment for PAC and identify therapeutic agents. We performed tumor-only next-generation sequencing (NGS) with targeted panels to explore the molecular changes underlying PAC patients in Taiwan. The Ion Torrent Oncomine Comprehensive Panel (OCP) was used for PAC metastatic lesions, and more PAC samples were sequenced with the Ion AmpliSeq Cancer Hot Spot (CHP) v2 panel. Five formalin-fixed paraffin-embedded (FFPE) metastatic PAC specimens were successfully assayed with OCP, and *KRAS* was the most prevalent alteration, which might contraindicate the use of anti-EGFR therapy. One PAC patient harbored a *FGFR2* p. C382R mutation, which might benefit from FGFR tyrosine kinase inhibitors. An additional 38 samples assayed with CHP v2 showed 100 hotspot variants, collapsing to 54 COSMID IDs. The most frequently mutated genes were *TP53*, *KRAS*, and *PDGFRA* (29, 23, 10 hotspot variants), impacting 11, 23, and 10 PAC patients. Highly pathogenic variants, including COSM22413 (*PDGFRA*, FATHMM predicted score: 0.88), COSM520, COSM521, and COSM518 (*KRAS*, FATHMM predicted score: 0.98), were reported. By using NGS with targeted panels, somatic mutations with therapeutic potential were identified. The combination of clinical and genetic information is useful for decision making and precise selection of targeted medicine.

## 1. Introduction

Pancreatic adenocarcinoma (PAC) is the 8th leading cause of cancer-related deaths in Taiwan, and its incidence is increasing [1]. Most PAC patients are diagnosed when the tumor is relatively large and has extended beyond the pancreas. There are several reasons for this delay. First, because of its anatomic location, the pancreas is not easily accessible with conventional diagnostic imaging tools. Second, initial symptoms of PAC are usually unremarkable, and clinical workup is often procrastinated until the onset of more suspicious signs. Third, cystic precursor lesions of PAC are not easily distinguishable from benign cysts and may represent a diagnostic dilemma that eventually delays the correct diagnosis [2]. PAC is often diagnosed at late stages because patients are often asymptomatic, so having more validated genetic biomarkers can augment early diagnosis and proper treatment. In addition, PAC is the most lethal human malignancy, with a dismal 5-year overall survival rate of less than 5%. Even with resectable tumors, the 5-year survival rate never exceeds 15%. It is estimated that 10% of PACs show familial aggregation consistent with genetic susceptibilities. However, in most instances, the genetic basis for hereditary PAC has not yet been identified [3,4]. The development of PAC involves successive accumulation of multiple genetic alterations with significant heterogeneity. Understanding the molecular pathogenesis and heterogeneity of PAC may facilitate personalized treatment for PAC and yield potential therapeutic targets [5,6,7].

Previous comprehensive exome sequencing of PAC revealed that dozens of alterations accumulated in each cancer, while most were rare/private mutations and were passengers by themselves. However, these studies also identified a number of recurrent aberrations, such as driver mutations that played a critical role during carcinogenesis, involving at least 12 cellular pathways implicated in PAC development [8,9,10]. However, all of these studies were conducted in Western countries, and no study has assessed the molecular alterations of patients with PAC in Taiwan. From past experiences, the patterns of driving mutations might be very diverse across different ethnic groups [11,12]. Therefore, we used state-of-the-art next-generation sequencing (NGS) with targeted panels to explore the molecular alterations underlying PAC in Taiwan. The study aimed to identify genetic alterations that might be targetable with existing drugs or serve as biomarkers. PAC is among the most malignant neoplasms, while research on PAC relies on clinical, pathologic, and molecular features for biomarker discovery and corresponding treatment. This study aimed to decipher genetic aberrations in Taiwanese patients with PAC.

## 2. Results

### 2.1. Part I: Retrospective Cohort with the OCP

Six FFPE PAC specimens were retrieved and tested in part I (retrospective cohort, Figure 1) of the study, five of which had adequate DNA/RNA for the Oncomine Comprehensive Panel v1 (OCP, Thermo Fisher Scientific, Waltham, MA, USA). These metastatic samples were from deceased PAC patients, including two from lymph nodes, two from malignant effusion, and one from liver metastasis. Approximately 150 to 200 unfiltered variants of different types were found in each sample (Table 1), many of which were probable passengers. Most Taiwanese patients with PAC harbored *KRAS* mutations, as previous studies have shown [13]. There was one *FGFR2* mutant case. The distribution of CNVs found in this study is detailed in Figure 2. Variant calling using the Partek Flow software with SAM tools (Partek Incorporated, St. Louis, MO, USA) was used to draw Sankey diagrams of the five assayed samples (Figure 3). Appendix A shows a variant impact heatmap of 5 PAC samples assayed with the OCP. At least one actionable mutation was reported for each case (Table 1).

### 2.2. Part II: Prospective Cohort with the CHP

A total of 39 archived FFPE samples of PACs were enrolled in part II (prospective cohort, Figure 1). Initially, both the quantity and quality of DNA extracted were unsatisfactory, and a quality improvement program was arranged; the REPLI-g (Qiagen, Hilden, Germany) and GenomePlex (Sigma, part of Merck KGaA, Darmstadt, Germany) kits were adopted with enhanced DNA yield from FFPE samples evidenced by the Qubit fluorimeter (Thermo Fisher Scientific, Waltham, MA, USA). Finally, a total of 38 PAC samples were successfully sequenced by the Ion AmpliSeq Cancer Hot Spot v2 panel (CHP, Thermo Fisher Scientific, Waltham, MA, USA) after excluding one heavily degraded sample (Appendix A for clinical and histological features).

The results of the Ion Reporter variant caller identified 1008 unfiltered variants from 38 patients (0 variant from one PAC patient, P11). The range was between 15 and 56 variants per sample, with a median of 24. The number of impacted genes was 41 (variants per gene: 1–152 variants/gene). The annotation sources were 100 hotspots (collapsed to 54 COSMIC IDs) and 895 novel/unknown IDs. Among filtered hotspot regions, 34 out of 38 (89%) patients reported at least 1 hotspot alteration, with a median of 3 per patient (range: 1–8). Frequently impacted genes and the number of associated alterations is detailed in Table 2 (Appendix A shows the tabulation of individual PAC subjects with impacted genes). Counted by impacted subject/variant, the most common mutations came from *KRAS* (23 samples/variants), *TP53* (11 samples/29 variants), and *PDGFRA* (10 samples/variants). Up to 5 variants could be detected within each patient with PAC.

Frequently impacted genes were *KRAS* (23 samples), *TP53* (11 samples), and *PDGFRA* (10 samples, Table 2). The COSMIC database was consulted for functional annotations of actionable mutations (Appendix A), and recurrent pathogenic variants are detailed in Appendix A, including highly pathogenic variants COSM22413 (*PDGFRA*, FATHMM predicted score: 0.88), COSM520, COSM521, and COSM518 (*KRAS*, FATHMM predicted score: 0.98). Appendix A shows variant impact heatmaps of Taiwanese patients with PAC assayed with CHP, while mucinous PACs and PACs with pancreatic intraepithelial neoplasm (PanIN) precursors are depicted separately; roughly comparable distributions of genetic alterations in *TP53*, *APC*, *SMAD4*, *PTEN*, *PIK3CA*, and *CDKN2A* were observed.

### 2.3. Mutational Landscape of Taiwanese Patients with PAC

To further integrate the findings from both parts of the study, Figure 4 and Figure 5 show the OncoPrinter plots of 43 Taiwanese patients with PAC with and without excluding germline mutations and alterations of unknown significance. Appendix A show the corresponding OncoPrinter plots from 176 PACs from The Cancer Genome Atlas (TCGA) Firehose Legacy after excluding 8 cases of pancreatic neuroendocrine tumors [14,15]. The most prevalent mutations (more than 50% of the study cohort, after excluding germline and unknown alterations) came from *KRAS* (74%), followed by *KDR* (59%) and *TP53* (56%, Figure 5). Figure 6 shows MutationMap plots of *KRAS*, *TP53, HRAS*, *PDGFRA,* and *FGFR2*. Hotspot mutation of *KRAS* G12D was prominent, while the distribution of *TP53* mutations was much more even, except for *P72R*. The *HRAS* G13D mutation was likely oncogenic, while functional impact of *PDGFRA* mutations remained unknown. Although not fully investigated, *FGFR2* C382S was considered likely oncogenic as C382R, which had been annotated as such.

### 2.4. KRAS Mutations Were Most Prevalent in Taiwanese Patients with PAC

Based on the results from both retrospective and prospective cohort, *KRAS* mutations were the most prevalent among Taiwanese patients with PAC, as 4 out of the 5 OCP assays reported *KRAS* mutations (p.G12D, p.G12R, and two p.G12V) and 23 out of 38 Taiwanese patients with PAC. Collectively, 74% of 43 assayed samples harbored a variant in *KRAS*, and a G12D hotspot was identified. Among 176 TCGA samples with PAC, the prevalence of *KRAS* mutation after excluding germline variants was 65%.

## 3. Discussion

Long-term survival of PAC remains stubbornly low, and there is an unmet need for early detection and efficient systemic treatment. Although treatment outcomes for many types of cancer have improved, PAC survival has lagged significantly behind. One major limitation comes from very few treatment options for PAC. A better understanding of PAC may lead to new treatment options and improved clinical outcomes for this lethal disease [16]. In the current study, we hypothesized that by using NGS, genetic alterations guiding the selection of targeted therapies could be identified. By predefining a set of relevant somatic alterations, targeted panels identified variants that could be linked to potential therapeutic strategies [17,18]. Two commercialized panels, namely, the OCP and CHP, were adopted to fulfill this purpose. The OCP was designed for compatibility with routine FFPE tissues, augmenting its clinical applicability. The CHP provided a more cost-effective and scalable solution for routine practice.

Initially, six FFPE metastatic samples of Taiwanese patients with PAC were evaluated with DNA/RNA extraction, and five with adequate nucleic acids were sequenced by the OCP. Up to 150 and 200 unfiltered alterations were found in each sample, while most were probable bystanders with no relevant therapy. In part I of the study, there was at least one actionable mutation found in each PAC patient, making personalized therapy possible. These actionable mutations corresponded to potentially matched treatments, which could be the targets of novel therapeutics.

The presence of *KRAS* mutations may be a predictive biomarker against the use of anti-EGFR antibodies. In lung cancer treatment, *KRAS* is downstream of the EGFR pathway; therefore, tyrosine kinase-based treatment with gefitinib and erlotinib is ineffective when *KRAS* is constitutively activated [19,20]. However, many clinical trials that combine other tyrosine kinase inhibitors (TKIs) and chemotherapy are ongoing for these *KRAS* mutant PACs [21]. In addition, knockdown of mutant *KRAS* with RNA interference may be a potential therapeutic strategy in the near future [22]. Recently, a subset of *KRAS* wild-type young PAC has been identified, but the recognition of alternative oncogenic drivers that are also targetable is urgently needed, further highlighting the importance of *KRAS* alterations [23]. There was one *FGFR2* mutant case (FJU01), and the NGS results showed the possibility of FGFR TKI treatment in the future, as there are several FGFR TKI trials with potential therapeutics, such as BGJ-398, ponatinib, TAS-120, alpelisib + BGJ-398, ARQ-087, BAY-1163877, FF-284, and JNJ-42756493 [24,25]. Although RNA sequencing showed no fusion gene in the current study, there is increasing evidence suggesting that fusion oncogenes are present not only in sarcoma but also in carcinoma [26]. This kind of alteration could also be a potential biomarker for targeted therapy, such as crizotinib for treating lung cancer with EML4-ALK translocation [27]. Many CNVs were also found in this study, but their meaning needs further investigation. Further studies to clarify whether the gain of oncogenes or loss of suppressors represent prognostic or predictive biomarkers for PAC are warranted.

The CHP was adopted for part II of the study with more samples assayed. The choice of the CHP rather than OCP was based on economic considerations. The CHP v2 surveyed hotspot regions covering 50 oncogenes and tumor suppressor genes, with wide coverage of *KRAS*, *BRAF,* and *EGFR* genes, which were evidenced as being PAC-relevant from part I of the study. Notably, it is also the CHP platform that was adopted in the NCI MATCH trial [28]. An example is the COSMIC 518 *KRAS* mutation involved in the MAPK, EGFR1, IL2, IL3, IL5, and ErbB pathways, which displayed a high pathogenic score. Although the COSMIC database identified *KRAS*, *PFGFRA,* and *KIT* as being pathogenic (Appendix A), only *KRAS* was predictive in terms of matched alteration-drug combinations. A mutant *KRAS* gene is a biomarker for many cancer types, as this gene has controlled cell cycle division and cancer cell growth. It is also convenient to conduct summarized cohort and subgroup analysis by the method of variant impact heatmaps, as a moderate sample size of part II made whole cohort and subgroup clustering analyses possible. A variant impact heatmap of PACs with mucinous and PanIN precursors showed roughly comparable distributions of genetic alterations in *TP53*, *APC*, *SMAD4*, *PTEN*, *PIK3CA,* and *CDKN2A* (Appendix A).

Although Ryan et al., highlighted that more than 90% of PACs were associated with activating *KRAS* mutations, the frequency was much higher for intraepithelial neoplasms (>90%) than for mucinous neoplasms (40–65%). For tumor suppressors such as *CDKN2A*, *TP53,* and *SMAD4*, the aberrant rate increased with a higher nuclear grade, albeit alterations in tumor suppressor genes rarely led to targeted therapy. The proposed *GNAS* oncogenic mutations were not observed in PAC with mucinous precursors in the current study [21]. Figure 6 shows that the most prevalent mutations (more than 50% of the study cohort, after excluding germline and unknown alterations) came from *KRAS* (74%), followed by *KDR* (59%) and *TP53* (56%), while only *KRAS* (65%) and *TP53* (60%) were prevalent among more than half of 176 cases from the TCGA cohort. The much lower *KDR* mutations in the latter might be related to assay-specific discrepancies, while true ethnic discrepancies could not be totally ruled out. On the other hand, a much higher rate of *KDR* mutations of our cohort could be a novel finding or perhaps, is a sequencing or bioinformatics (really germline) error. Sanger sequencing of tumor and paired normal tissue would be possible in future study.

Singhi et al., conducted by far the largest study on approximately 3600 samples of PAC and found that the most frequently mutated genes, i.e., *KRAS*, *TP53*, *CDKN21*, and *SMAD4* cannot be targeted with readily available existing drugs [29]. In 2020, the FDA approved olaparib for maintenance treatment of germline BRCA-mutated metastatic PAC whose disease has not progressed on at least 16 weeks of first-line platinum-based chemotherapy based on a phase III POLO trial [30]. The most common pathogenic germline mutations are in *BRCA1*, *BRCA2*, and *ATM* and, more rarely, in *PALB2*, *MLH1*, *MSH2*, *MSH6*, *PMS2*, *CDKN2A*, and *TP53*, among others, for an aggregate frequency of 3.8% to 9.7% [31,32]. Although only somatic alterations were investigated in the current study, several aberrant DNA damage repair genes were reported from Taiwanese patients with PAC, and reflux germline testing should be indicated based on alterations in tumor-only sequencing.

There were some limitations of the study. First, the modest sample size might hamper externalization of sequencing results. The 50 targeted genes of the CHP were covered by the OCP, so there was no concern of comparability. In addition to the limited sample size, it should be noted that only 15% of PACs were resectable at the time of diagnosis, which means that the majority of PAC genetic analyses were conducted on an early-stage disease [33,34]. In the current study, part I samples were derived from metastatic lesions, while for part II, all specimens came from primary pancreatic neoplastic tissue in an effort to broaden clinical scenarios of PAC [35]. Subgroup analyses between PAC with mucinous and PanIN precursors might be differential if more samples were enrolled. Second, although PAC is characterized by diverse, large chromosomal changes with forms of amplifications, deletions, and rearrangement, only the five OCP assays investigating CNVs and structural aberrations of the 38 CHP assays were left undetermined. Third, as PAC is hard to diagnose due to the difficulty in obtaining tumor samples from patients, the feasibility and prognostic value of circulating tumor DNA (ctDNA) in PAC is being tested rigorously [36]. The knowledge learned from the current study and other genetic studies may pave the way for future circulating biomarkers to screen, guide treatment, and monitor the disease progress of PAC.

## 4. Materials and Methods

In the current study, tumor-only NGS was performed to decipher genetic alterations in Taiwanese patients with PAC. The whole study protocol was reviewed and approved by the IRB of Cathay General Hospital; informed consent of part I (retrospective cohort) was waived, while signed informed consent was obtained from all participants of part II (prospective cohort). All methods were performed in accordance with the relevant guidelines and regulations.

### 4.1. Study Population

The enrollment criteria were adult patients (more than 20 years old) with pathology-confirmed PAC. Neuroendocrine tumors with pancreatic origin were excluded. In the first part (retrospective cohort), we took advantage of readily available formalin-fixed paraffin-embedded (FFPE) specimens of metastatic lesions from deceased patients with PAC to evaluate the feasibility of extracting nucleic acids for targeted sequencing (enrollment period: from March 2011 and August 2013). The OCP v1 was used as the initial screening tool for detecting tumor genetic alterations and could be used as a candidate for downstream analyses. In the second part (prospective cohort), more samples were collected to elucidate the oncogenesis of Taiwanese patients with PAC using the CHP v2 panel (enrolled between November 2013 and August 2016). Figure 1 shows the whole study workflow. Planned sample size was 6 for part I (retrospective cohort) and 39 for part II (prospective cohort), with a total of 45 Taiwanese patients with PAC. For part I, readily available archived FFPE samples from deceased patients with PAC were the source of assayed samples, while for part II, prospective enrollment was carried out for Taiwanese patients with PAC.

### 4.2. Nucleic Acid Extraction

The source of nucleic acids for targeted sequencing was stored pathological slides or paraffin blocks. Extraction of DNA and RNA was performed according to laboratory manuals; hematoxylin and eosin (H&E)-stained slides were reviewed by one certified pathologist (CYL) to ascertain the presence of adequate PAC cells. Paraffin blocks or slides with cancer cells less than 70% of the section area were excluded, while paraffin was removed by xylene extraction and then by ethanol washes. Nucleic acid was extracted from 5 × 5 μm sections with the QIAmp DNA FFPE Tissue Kit or AllPrep DNA/RNA FFPE Kit (Qiagen, Valencia, CA, USA), while the quality and concentration of harvested DNA/RNA was determined by the Qubit fluorimeter (Invitrogen, part of Thermo Fisher Scientific, Waltham, MA, USA) augmented by the Qubit dsDNA HS (High Sensitivity) and Qubit dsDNA BR (Broad Range) Assay Kits (Thermo Fisher Scientific, Waltham, MA, USA). In addition, PCR of GAPDH fragments was used to determine whether the degree of fragmentation was acceptable for amplification and sequencing. A quality improvement with a protocol amendment was made when unsatisfactory nucleic acid extraction was encountered for the CHP (see below).

### 4.3. Targeted Sequencing Panel

For the OCP v1 assay, 143 preselected genes were designed to interrogate somatic mutations, including single/multinucleotide variants (SNVs/MNVs), insertions/deletions (INDELs, 73 genes), and copy number variations (CNVs), including gains (49 genes) and losses (26 genes), recognized as oncogenes or tumor suppressors recurrently altered in solid tumors [37]. In addition to DNA, RNA was extracted to test 22 preselected fusion genes. The CHP v2 assay was designed to amplify 207 amplicons covering approximately 2800 COSMIC mutations from 50 oncogenes and tumor suppressor genes [38,39].

### 4.4. NGS Experiments

Amplicon libraries were constructed with multiplex PCR primers following the manufacturer’s instructions using a total of 10 ng of DNA (20 ng of DNA and 15 ng of RNA input for the OCP) per sample with the Ion AmpliSeq Library Kit v2. Templates were generated with the Ion Chef or Ion One Touch 2 system, while sequencing was performed on an Ion 318 v2 chip with the Ion PGM Hi-Q Sequencing Kit (all from Thermo Fisher Scientific, Waltham, MA, USA). The Ion Torrent Personal Genome Machine (Ion PGM) operated by Yourgene Health (New Taipei, Taiwan, for the OCP) and Fu-Jen Catholic University (for the CHP) was the platform for NGS.

### 4.5. Data Analysis

Torrent Suite and Ion Reporter software (both from Thermo Fisher Scientific, Waltham, MA, USA) were used for data analysis. Raw data processing, alignment, and variant calling were performed with Torrent Suite software, and variants were called with the Torrent Variant Caller plug-in, followed by downstream analyses by Ion Reporter software with the workflow “Oncomine Comprehensive v2.0—DNA -Single Sample r.0” selected and filter chain “Oncomine Variants” applied. The reference genome was hg19, and a cutoff of 500× coverage was chosen. Partek Flow software (Partek Incorporated, St. Louis, MO, USA) was used as an additional calling algorithm [40]. MutationMapper and OncoPrinter from the cBioPortal Cancer Genomics were used for visualization purposes [14,15]. Additional annotations and drug-alteration matching were performed by OncoKB [41].

## 5. Conclusions

By using NGS with a targeted panel, somatic mutations with therapeutic potential were identified. The combination of clinical and genetic information is useful for decision making and precise selection of targeted medicine. There were some discrepancies in mutational landscapes between PAC patients of Taiwan and TCGA cohort, while *RKAS* remained the most prevalent actionable mutation. Although *KRAS* mutations play a major role during PAC tumorigenesis, there remains an unmet need for effective treatment for this lethal disease and more novel therapeutics are eagerly needed if NGS could identify valid and corresponding biomarkers. Further studies to take advantage of NGS to decipher genetic alterations underpinning PAC and to identify corresponding therapeutics are warranted.

## Figures and Tables

**Figure 1 ijms-23-01579-f001:**
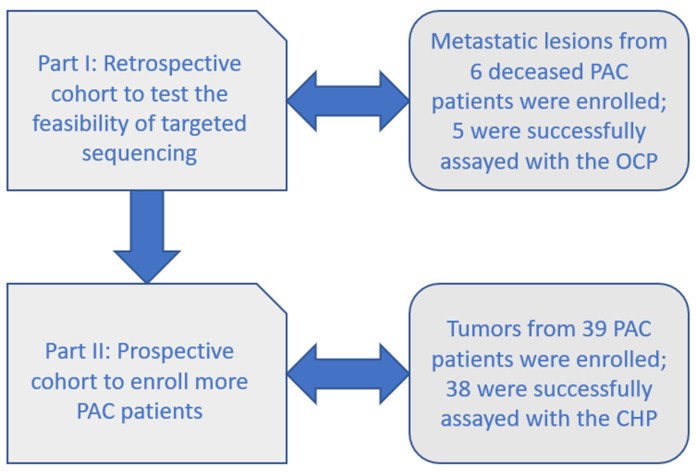
Study workflow (PAC: pancreatic adenocarcinoma, OCP: Oncomine Comprehensive Panel, CHP: Cancer Hotspot Panel).

**Figure 2 ijms-23-01579-f002:**
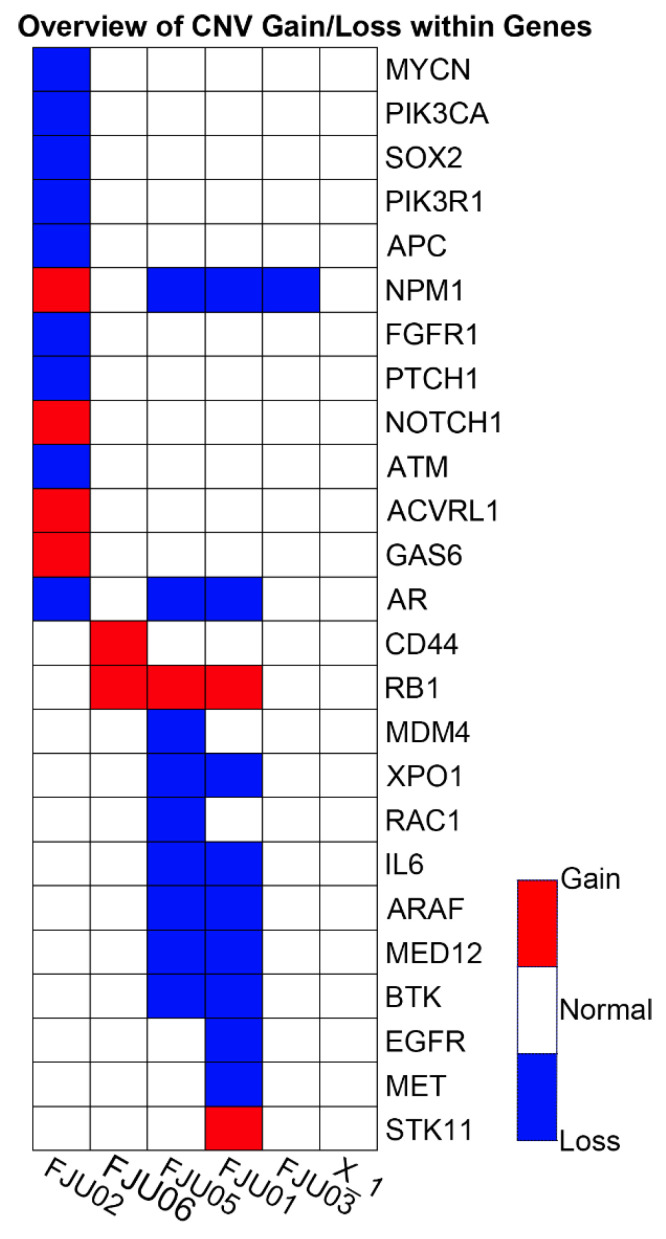
The occurrence of CNVs within genes among five Taiwanese patients with PAC (CNV: copy number variation, PAC: pancreatic adenocarcinoma).

**Figure 3 ijms-23-01579-f003:**
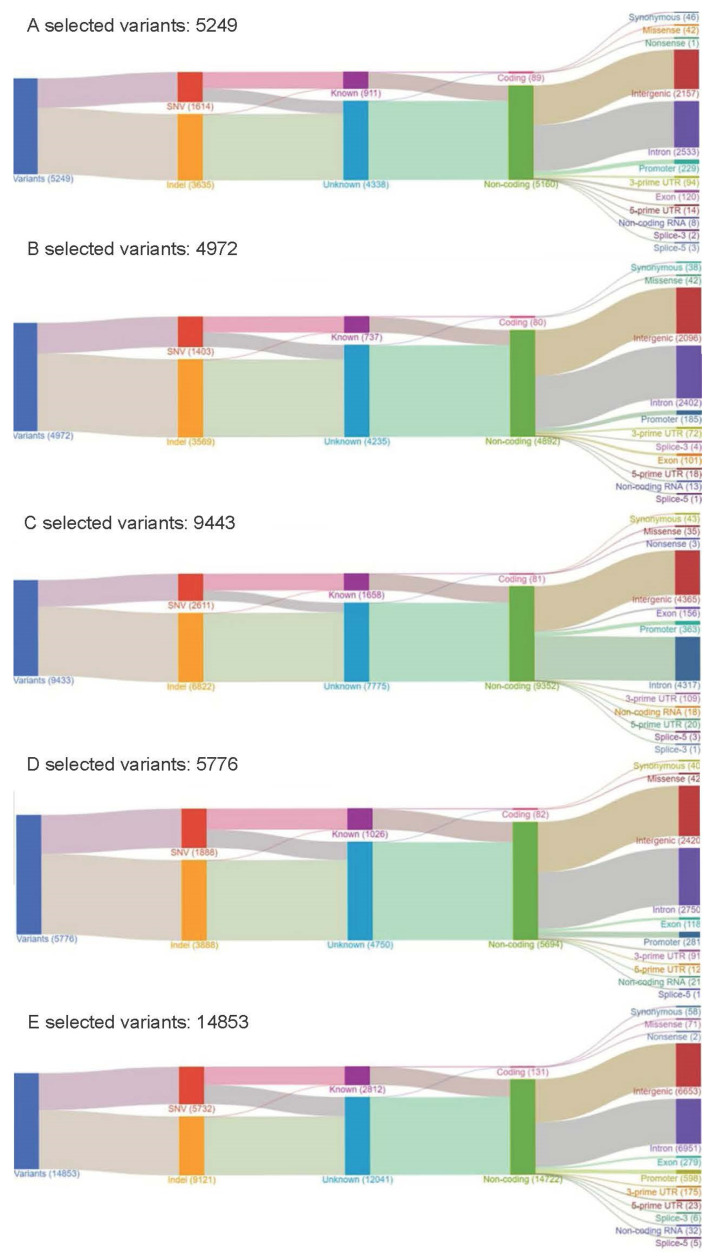
Sankey diagrams of five Taiwanese patients with PAC assayed with OCP (PAC: pancreatic adenocarcinoma, OCP: Oncomine Comprehensive Panel).

**Figure 4 ijms-23-01579-f004:**
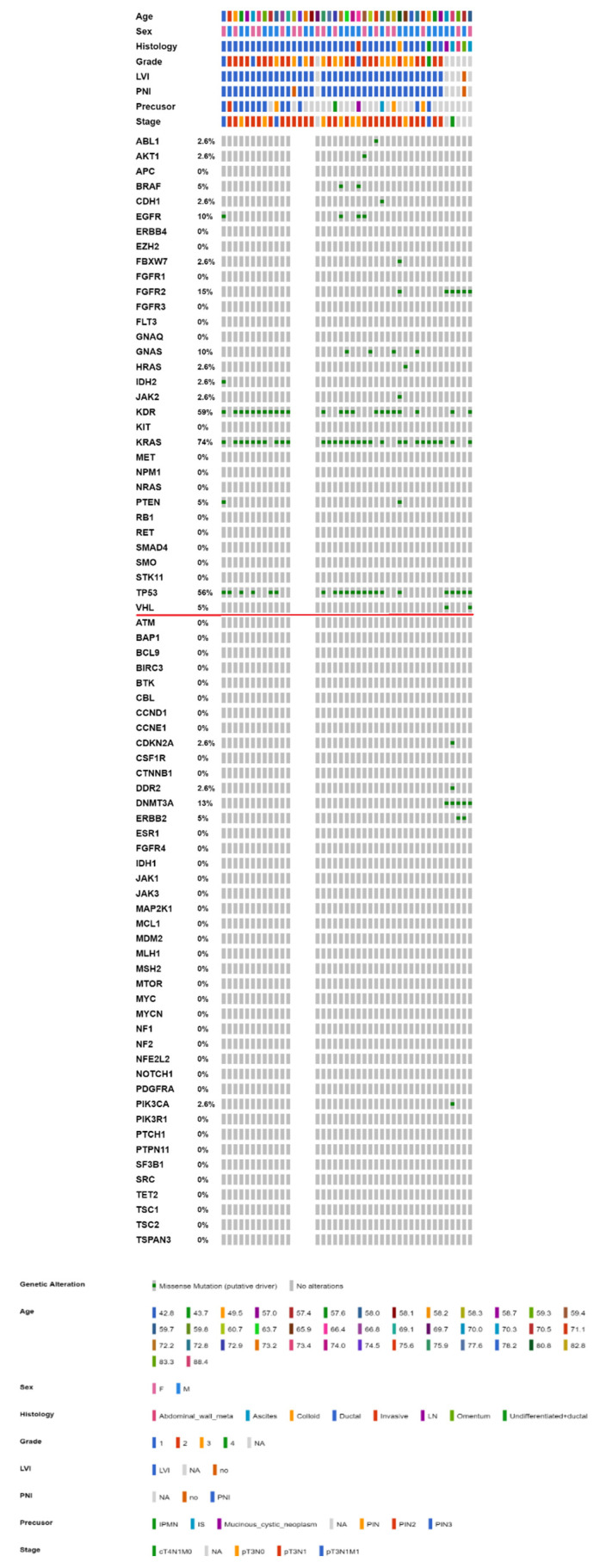
OncoPrinter of 43 Taiwanese patients with PAC (38 with the CHP and 5 with the OCP). Genes above the horizontal red line are common genes across both platforms, while those below are interrogated by OCP only (PAC: pancreatic adenocarcinoma, OCP: Oncomine Comprehensive Panel, CHP: Cancer Hotspot Panel).

**Figure 5 ijms-23-01579-f005:**
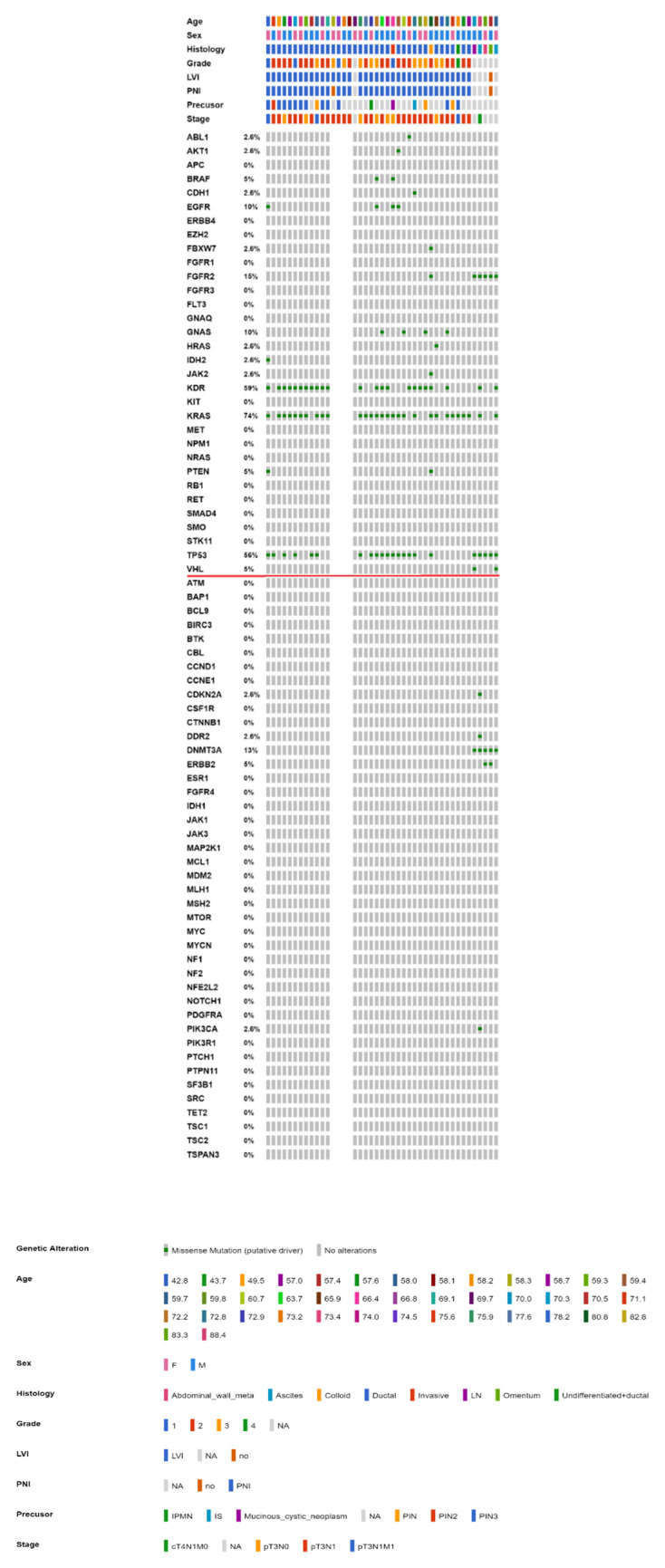
OncoPrinter of 43 Taiwanese patients with PAC (38 with the CHP and 5 with the OCP) with germline mutations and alterations of unknown significance excluded. Genes above the horizontal red line are common genes across both platforms, while those below are interrogated by OCP only (PAC: pancreatic adenocarcinoma, OCP: Oncomine Comprehensive Panel, CHP: Cancer Hotspot Panel).

**Figure 6 ijms-23-01579-f006:**
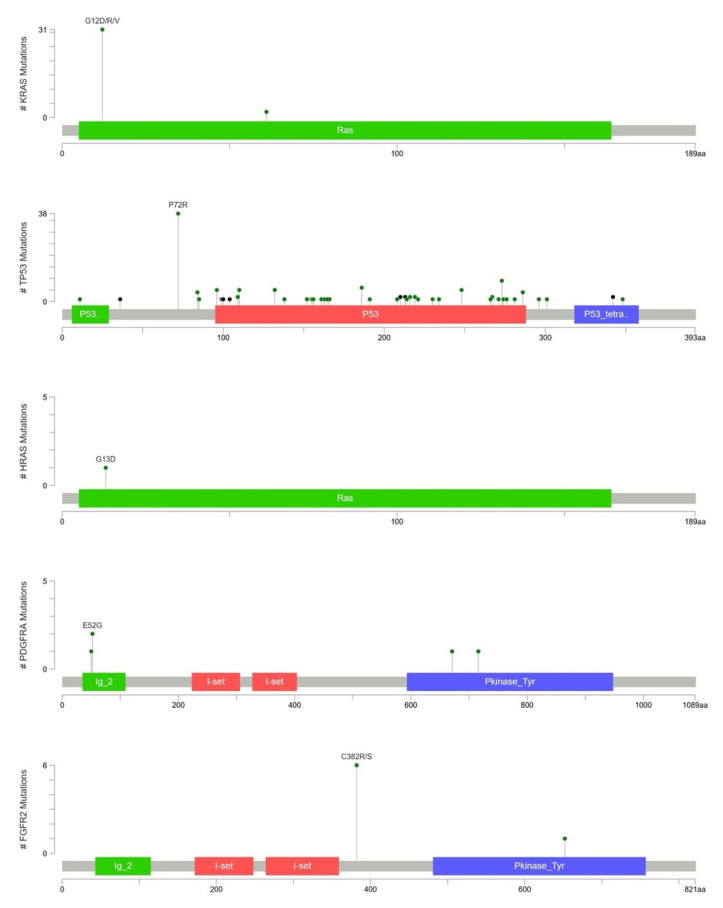
MutationMap of *KRAS*, *TP53*, *HRAS*, *PDGFRA,* and *FGFR2*.

**Table 1 ijms-23-01579-t001:** Summary of variants reported from five Taiwanese patients with PAC assayed with the OCP (PAC: pancreatic adenocarcinoma, OCP: Oncomine Comprehensive Panel, SNV: single nucleotide variant, MNV: multi-nucleotide variant, CNV: copy number variation, INDEL: insertion or deletion, COSMIC: the Catalogue of Somatic Mutations In Cancer).

Sample ID/Tumor Source	No. of SNVs/MNVs	No. of INDELs	No. of CNVs	No. of Fusions	Total Positive Variants Including SNVs/MNVs/CNVs/Fusions	No. of SNVs/MNVs/INDELs with COSMIC IDs	No. of Non-Synonymous Variants	Actionable Mutations
FJU01/Metastatic lymph node	154	19	11	-	184	1	59	*FGFR2* p.C382R
FJU02/Malignant effusion	179	4	13	-	196	3	83	*KRAS* p.G12R/
								*ATM* deletion
FJU03/Metastatic lymph node	137	10	1	-	148	2	44	*KRAS* p.G12V/
								*TP53*
								p.R273C
FJU05/Omentum metastasis	136	13	10	-	159	1	43	*KRAS*
								p.G12D
FJU06/Malignant effusion	135	10	2	-	147	2	44	*KRAS*
								p.G12V/
								*TP53*
								p.248Q

**Table 2 ijms-23-01579-t002:** Frequently impacted genes and the number of associated variants among 100 hotspot regions from Taiwanese patients with PAC assayed with the CHP (PAC: pancreatic adenocarcinoma, CHP: Cancer Hotspot Panel).

Gene Symbol	No. of Variants	No. of PAC Samples
*TP53*	29	11
*KRAS*	23	23
*PDGFRA*	10	10
*KIT*	6	6
*PTEN*	6	4
*SMARCB1*	6	6
*GNAS*	4	2
*MET*	4	4
*CDKN2A*	2	2
*CTNNB1*	2	2
*EGFR*	2	2
*IDH1*	2	2
*STK11*	2	2
*SMO*	1	1
*HRAS*	1	1

## Data Availability

Genomic data of the study were secured by the primary investigators as requested by IRB and might be available in an anonymous manner upon reasonable request.

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
