# Peer review of "Deciphering Genetic Alterations of Taiwanese Patients with Pancreatic Adenocarcinoma through Targeted Sequencing"

_ijms, 2022, doi:10.3390/ijms23031579_

Round 1
Reviewer 1 Report
In this study, Huang et al. investigated the genetic alterations in Taiwanese patients with pancreatic adenocarcinoma (PAC). Using targeted sequencing, they explored genetic aberrations in a small retrospective cohort and a larger prospective cohort and identified alterations of clinical relevance in PAC. Overall, the study provides novel insights into the genetic alterations of PAC in this population. The study would benefit from these comments:
- The Results section is very brief. Expanding this section and highlighting the most important findings is needed. Perhaps the results highlighted in the Discussion section need to be transferred to the Results section. It is also important to use descriptive titles for sub-sections in the Results section; for example, “KRAS mutations are prevalent in PAC tumors from Taiwanese patients”. This is important to highlight key findings and guide the reader through the manuscript.
- Figure 6 is misplaced after Figure 1. I would suggest relocating Fig. 6 to serve as Fig. 1 as it summarizes the design of the study
- I would not recommend using the term “Taiwanese pancreatic adenocarcinoma (PAC)”. Perhaps, “Taiwanese patients with pancreatic adenocarcinoma” is more appropriate. Thus, I suggest rephrasing Taiwanese throughout the manuscript
Author Response
- The Results section is very brief. Expanding this section and highlighting the most important findings is needed. Perhaps the results highlighted in the Discussion section need to be transferred to the Results section.
Response: We thank Reviewer’s comment and expand the Results by transferring some materials from the Discussion during revision.
- It is also important to use descriptive titles for sub-sections in the Results section; for example, “KRAS mutations are prevalent in PAC tumors from Taiwanese patients”. This is important to highlight key findings and guide the reader through the manuscript.
Response: As both retrospective (part I) and prospective cohort (part II) reported that KRAS mutations were the most prevalent alterations among Taiwanese patients with PAC, we add a new subsection 2.4 to highlight this key finding.
- Figure 6 is misplaced after Figure 1. I would suggest relocating Fig. 6 to serve as Fig. 1 as it summarizes the design of the study
Response: We have re-numbered Figure 6 (original) as Figure 1 (revised) and successive Figures accordingly.
- I would not recommend using the term “Taiwanese pancreatic adenocarcinoma (PAC)”. Perhaps, “Taiwanese patients with pancreatic adenocarcinoma” is more appropriate. Thus, I suggest rephrasing Taiwanese throughout the manuscript.
Response: We have corrected this point throughout the revised manuscript.
Reviewer 2 Report
Dear Authors:
The authors have carried out the study “Deciphering genetic alterations of Taiwanese pancreatic adenocarcinoma with targeted sequencing”. The aim of this study is to explore the molecular alterations underlying Pancreatic adenocarcinoma (PAC) in Taiwan for potential clinical implications.
The authors have conducted a comprehensive, rigorous, and scientifically correct study owith obvious implications for daily clinical practice.
Nevertheles, some considerations need to be taken into account:
- The data collection period is not identified in the study. In what period of time was the study carried out? What was the study period?
- The sample size is small. Only 5 patients were collected in part I of the study and 38 in part II, which seems scarce and could limit the development of robust conclusions.
- Conclusions are scant, predictable, and with limited scientific soundness. They can be extrapolated to any other tumor since they do not make any mention of the object of the study
- Bibliographic references should be reflected homogeneously. Some quote the doi and some don't as an example.
The manuscript requires a reorganization of the information that makes reading attractive when you get to the conclusions.
Kind regards
Author Response
- The data collection period is not identified in the study. In what period of time was the study carried out? What was the study period?
Response: Enrollment period for both part I and part II has been added to revised section 4.1.
- The sample size is small. Only 5 patients were collected in part I of the study and 38 in part II, which seems scarce and could limit the development of robust conclusions.
Response: Limited sample size a major concern of the study and has been addressed in the last paragraph of Discussion (Page 9, Line 333-335)
- Conclusions are scant, predictable, and with limited scientific soundness. They can be extrapolated to any other tumor since they do not make any mention of the object of the study.
Response: We have expanded Conclusions during revision: “There were some discrepancies in mutational landscapes between PAC patients of Taiwan and TCGA cohort, while RKAS remained the most prevalent actionable mutation. Although KRAS mutations play a major role during PAC tumorigenesis, there remains an unmet need for effective treatment for this lethal disease and more novel therapeutics are eagerly needed if NGS could identify valid and corresponding biomarkers. Further studies to take advantage of NGS to decipher genetic alterations underpinning PAC and to identify corresponding therapeutics are warranted”.
- Bibliographic references should be reflected homogeneously. Some quote the doi and some don't as an example.
Response: We have corrected this error.
Round 2
Reviewer 2 Report
Dear authors,
Thank you very much for reviewing, clarifying and modifying the allegations made previously. It is a very constructive work with high scientific rigor. I have communicated it to the editors of the journal.
Kind regards
Author Response
We thank Reviewer's comments.